# Restoring Functionalities in Chicken Breast Fillets with Spaghetti Meat Myopathy by Using Dairy Proteins Gels

**DOI:** 10.3390/gels8090558

**Published:** 2022-09-02

**Authors:** Chaoyue Wang, Leonardo Susta, Shai Barbut

**Affiliations:** 1Department of Food Science, Ontario Agricultural College, University of Guelph, 50 Stone Road East, Guelph, ON N1G 2W1, Canada; 2Department of Pathobiology, Ontario Veterinary College, University of Guelph, 50 Stone Road East, Guelph, ON N1G 2W1, Canada

**Keywords:** broiler myopathy, caseinate, meat texture, milk protein, spaghetti meat, whey protein

## Abstract

The use of caseinate, whole milk powder, and two whey protein preparations (WP; 2% *w*/*w*) was studied in minced meat made with normal breast (NB), and ones showing spaghetti meat (SM). SM is an emerging myopathy known for muscle fiber separation and lower protein content, costing $100s of millions to the industry. Using SM without dairy proteins resulted in a higher cooking loss (SM: 3.75%, NB: 2.29%; *p* < 0.05), and lower hardness (SM: 29.83 N, NB: 34.98 N), and chewiness (SM: 1.29, NB: 1.56) compared to NB. Using dairy proteins, except WP concentrate and WP isolate, significantly improved yield and increased hardness. Adding WP isolate to SM resulted in a similar texture profile as NB samples without dairy proteins (34 and 35 N hardness; 0.22 and 0.24 springiness; 1.57 and 1.59 chewiness values, respectively). Adding caseinate and whole milk to SM showed a more substantial effect of improving water-holding capacity, increasing hardness, gumminess, and chewiness compared to adding WP; i.e., adding caseinate and milk powder resulted in higher values for those parameters compared to NB without additives. Overall, it is shown that dairy proteins can be added to SM to produce minced poultry meat products with similar or higher yield and texture profiles compared to using normal breast fillets.

## 1. Introduction

The poultry industry utilizes non-meat additives, such as dairy proteins, carbohydrate gums, and starches to improve moisture retention and texture, and as fat replacements [1,2,3]. An emerging myopathy known as spaghetti meat (SM) has recently been causing significant challenges to the poultry meat industry [4]. While posing no food safety concern, SM fillets are ill-received by the market due to their soft texture and appearance, featuring detached fiber-like muscle [5]. That review found that 57% of consumers disliked fillets with myopathies, and scores were significantly lower with the increase in the severity of the myopathy. It is also reported that consumers from developed countries are increasingly concerned about how livestock are farmed; fillets with myopathies are deemed by certain consumers to be ‘abnormal muscle’ and are rejected based on appearance. Histological studies report degeneration, necrotic myofibers, and fragmentation of myofibers in the pectoralis major muscle. This is different from the prominent fibrosis in wooden breast (WB) meat, named after the hard tissue occurrence [6]. Reports on the causes of myopathies include muscle hypertrophy, hypoxia, mitochondrial and sarcoplasmic reticulum alterations, vascular damage, phlebitis, oxidative stress, and inflammatory processes [7].

Some economic losses could be recuperated by moving the meat to the secondary processing department. However, further processing by itself cannot mitigate textural deficiencies caused by myopathies [8]. For WB and white striping (WS), blade tenderization and marination can be employed to improve the texture of processed meat products [9]. However, for SM, current methods are ineffective in mitigating texture deficiencies and lower moisture retention caused by loose and fragmented muscle fiber. Deviation of other parameters, such as marinate uptake and pH, may also cause a deleterious effect to process formulations [10]. Therefore, finding an effective mitigation method to reduce/minimize the effect of SM is a priority for the poultry meat industry.

Studies have reported that adding dairy protein extracts or milk powder to chicken meat batters formed a filled gel with a reinforcing matrix [11]. In this system, added dairy proteins function as ‘fillers’ and occupy the voids inside the meat protein gel; thus, this increases the overall firmness. Spaghetti meat is reported to have lower protein content than normal fillets and showed a lower compression force after cooking [4]. It is hypothesized that without additives, meat batter using SM would have lower texture properties and water-holding capacity. Adding dairy proteins should improve the gel matrix of the meat batter. Therefore, this study aimed to evaluate the effects of different dairy proteins on improving the water binding and texture of minced meat made from chicken with SM myopathy, and compare it to samples without myopathy. Currently, there is no report on the effectiveness of dairy protein as additives to poultry products utilizing SM, featuring different proximate compositions compared to normal fillets.

## 2. Result and Discussion

Cooking losses were greater in the SM samples without dairy protein addition as compared to the NB samples (Figure 1). Overall, all the dairy protein additions, except WPC and WPI, helped reduce cooking loss. Similar results showed that adding various dairy proteins to NB chicken meat batters assisted in lowering cooking losses [12]. In the present study, we report on the additional contribution of dairy proteins to improve the water-holding capacity (WHC) of the SM samples. Adding caseinate or whole milk proteins to SM resulted in similar cooking losses as NB meat without added dairy proteins (Figure 1). Adding WPI significantly lowered the cooking loss of NB meat, while no significant effect was observed in SM. Others had also reported a greater improvement in WHC when caseinate was added to NB samples compared to WPC and WPI [13]. Jin et al. [14] evaluated the effect of caseinate on the physicochemical properties of sausages; they showed caseinate to significantly improve the cooking yield, shear force, and free moisture ratio to similar levels as other commonly used binders (e.g., soy protein isolate, egg white powder, pork plasma proteins). Kang et al. [15] reported that adding whole milk powder to chicken-breast emulsion-type sausage significantly improved the cooking yield, and minimized moisture and fat separation.

The hardness value for the NB sample (without dairy proteins) was greater (*p* < 0.05) than the value for the SM samples (34.98 vs. 29.83 N; Figure 2). This is because the SM meat already shows fragmentation of the muscle bundles (i.e., poor connective tissue structure). This also extends to the cooked state of the meat mass. In line with a previous study [11], dairy protein addition resulted in a significant increase in hardness. Caseinate and whole milk protein had a more significant effect than whey proteins on the SM samples. The ability to form a strong gel system was also observed when xanthan gums, soy proteins, or other binders were added to reduced-fat meat systems [16,17]. WPI and WPC added to the SM samples resulted in a hardness value of 33.93 N and 31.58 N, respectively. These values were similar to the NB samples without dairy protein addition. These values were comparable to a previous study where starches and carrageenan were added to low-fat chicken patties [18]. Zorbas et al. [19] added different levels of skim milk and whey powder; they reported a significant increase in apparent yield stress when whey powder was added to chicken or turkey meat emulsions. In the current study, SM showed significantly lower chewiness compared to the NB samples (Table 1).

Adding caseinate and whole milk powder to the NB and SM samples resulted in a significantly higher chewiness (e.g., SM: 1.29; SM + caseinate: 2.31; SM + whole milk powder: 2.81). A similar increase in hardness and chewiness was reported when whole milk powder was added to the chicken sausage [15]. Adding WPI or WPC to the SM samples resulted in values comparable to the NB samples without dairy protein addition. Youssef and Barbut [20] reported that dairy protein preparations interacted with the meat protein matrix and formed a synergistic matrix; thus, this improved the texture profile of meat batters. Lower effects on WHC and texture values were observed when dairy proteins, except for caseinate, were added to the SM samples. This could be caused by the difference in the composition of SM fillets (e.g., lower protein, higher fat content) and, hence, the ability to form a strong gel. Adding caseinate to the SM samples resulted in the highest increase in texture values and water-holding capacity. A clear difference in the samples could be observed between the SM samples with and without adding dairy proteins (Figure 3). Similar to the observations on whole fillets [21], minced meat made with SM did not significantly affect cohesiveness, springiness, and resilience (Table 1). In the present study, the effect of dairy protein on springiness and resilience was minimal; this was similar to earlier reports on the effects of dairy proteins on the texture of minced poultry meat [12,22]. Overall, all the proteins significantly increased the cohesiveness of the samples.

## 3. Conclusions

The SM samples showed a significantly higher cooking loss, lower hardness, gumminess, and chewiness compared to the NB samples. Caseinate or whole milk protein addition to SM resulted in a similar cooking loss as the NB samples. Adding whey protein did not significantly lower cooking loss, while adding WPI to SM resulted in texture values similar to NB without additives. Adding caseinate and whole milk significantly increased hardness and chewiness. Adding them to SM resulted in values higher than for the NB samples. Overall, whey proteins added to SM resulted in a similar texture profile as the NB samples without dairy protein addition. These findings showed that dairy proteins could be added to SM fillets to formulate a product with similar textural properties to normal fillets. Further studies could be conducted to find the optimized quantity and combination of dairy proteins to achieve the desired texture values.

## 4. Materials and Methods

### 4.1. Meat and Meat Batter Preparation

Normal breast fillets (NB) and SM fillets from broilers (average weight of 2.3 kg) were collected from three different flocks (one week apart) that were processed at a large local water chill poultry plant. Each sampling consisted of 15 NB and 15 SM fillets. Since SM is found at the cranial part of the chicken breast fillet, only the ‘fillets’ upper part was used. The cranial parts were cut and chopped using a food processor (Braun, model UK1-Type 4259, Kronberg, Germany) for 20 s. On each sampling day, the chopped NB and SM were allocated to five treatments: control (no dairy protein added); sodium caseinate (87 g/100 g protein, Herman Laue Spice Co., Uxbridge, ON, Canada); whole milk proteins (26 g/100 g protein, Herman Laue); whey protein concentrate (79 g/100 g protein, WPC-Herman Laue); and whey protein isolate (95 g/100 g protein, WPI-BiPro, Davisco International, Inc., Le Sueur, MN, USA). Each treatment consisted of 88 g minced meat mixed with 10 g water, 2 g sodium chloride, and either 2 g of caseinate, WPC, WPI, or 4 g of whole milk protein (the latter, because of its lower protein content). Salt and water were introduced to the meat samples (+ dairy proteins when applied) and mixed thoroughly, by hand, for 30 s. Duplicate 30 g samples were stuffed into 50 mL plastic test tubes and stored overnight at 4 °C. The sample preparation methods follow the procedure previously described to study the effects of adding dairy proteins to poultry meat batters [11,13,22].

### 4.2. Cooking and Cooking Loss

Samples in test tubes were cooked in a computer-controlled water bath (Thermo Haake, model W26, Newington, NH, USA) from 5 °C to an internal temperature of 72 °C within 1.25 h. The samples were then stored overnight at 4 °C; then, the loss liquid was measured and reported as cooking loss (free liquid divided by the initial meat mixture weight) [13,22].

### 4.3. Texture Profile Analysis (TPA)

The samples were cut into six pucks (20 mm diameter and 10 mm height) per treatment. TPA was performed using a texture analyzer (Stable Micro System TA.XT2, Texture Technologies Corp., Scarsdale, NY, USA), employing a flat cylindrical plate (10 cm diameter), descending at 1.5 mm/s and performing two cycle 50% compressions [13,22].

### 4.4. Statistical Analysis

The experiment was designed as a complete randomized block, with three separate trials (different flocks processed on different weeks). The treatments were set as independent effects, and the flocks were set as random effects. The one-way ANOVA option of the GLM procedure was performed using the SAS software package (SAS version 9.40; SAS Institute Inc., Cary, NC, USA). Tukey’s multiple comparisons were used to separate the means (*p*< 0.05) and were reported with their SEM.

## Figures and Tables

**Figure 1 gels-08-00558-f001:**
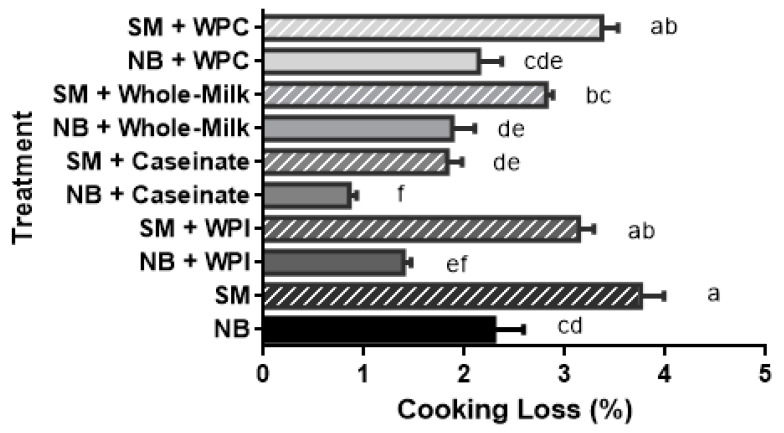
Cooking loss values for the minced normal breast (NB) chicken fillets and spaghetti meat (SM) samples, with and without dairy proteins (caseinate, whey protein concentrate, whey protein isolate, whole milk). ^a–f^ Means (*n* = 18) followed by a different letter are significantly different (*p* < 0.05). The stripes indicate SM samples; the bars show standard error.

**Figure 2 gels-08-00558-f002:**
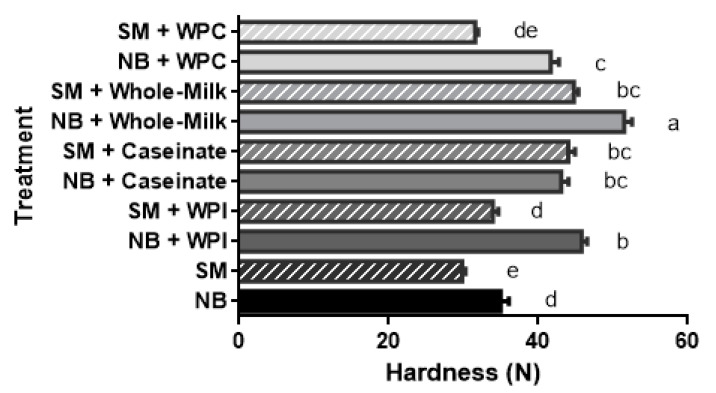
Hardness values for the minced normal breast (NB) chicken fillets and spaghetti meat (SM) samples, with and without dairy proteins (caseinate, whey protein concentrate, whey protein isolate, whole milk). ^a–e^ Means (*n* = 18) followed by a different letter are significantly different (*p* < 0.05). The stripes indicate SM samples; the bars show standard error.

**Figure 3 gels-08-00558-f003:**
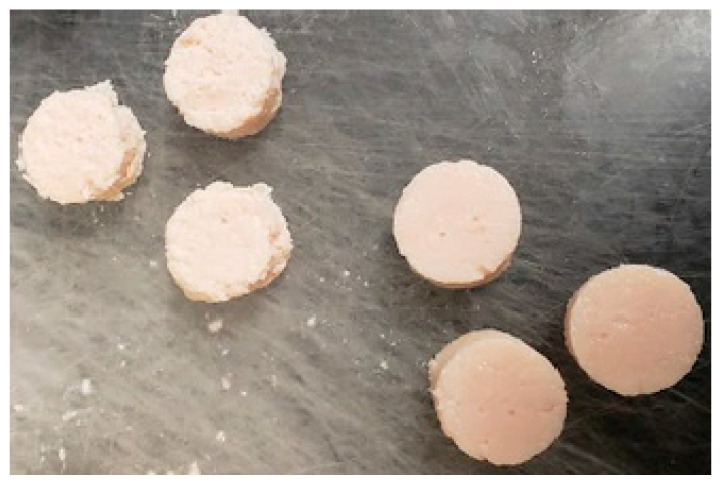
Sections of the cooked meat made with minced SM poultry breast meat (**left**: SM, **right**: SM + caseinate).

**Table 1 gels-08-00558-t001:** Effects of the dairy protein treatments on the normal breast (NB) and spaghetti meat (SM) samples, with and without dairy proteins (caseinate, whey protein concentrate, whey protein isolate, whole milk).

	TPA			
Treatment	Springiness	Cohesiveness	Chewiness	Resilience
NB	0.22 ± 0.01 ^e^	0.19 ± 0.01 ^de^	1.57 ± 0.14 ^ef^	0.05 ± 0.01 ^d^
SM	0.23 ± 0.01 ^de^	0.19 ± 0.01 ^e^	1.29 ± 0.06 ^f^	0.05 ± 0.01 ^cd^
NB + Whey protein isolate	0.25 ± 0.01 ^cd^	0.22 ± 0.01 ^b^	2.51 ± 0.11 ^bc^	0.06 ± 0.01 ^bc^
SM + Whey protein isolate	0.24 ± 0.01 ^cde^	0.20 ± 0.01 ^cde^	1.59 ± 0.09 ^ef^	0.05 ± 0.01 ^bcd^
NB + Caseinate	0.24 ± 0.01 ^cde^	0.21 ± 0.01 ^bcd^	2.2 ± 0.11 ^cd^	0.06 ± 0.01 ^bcd^
SM + Caseinate	0.24 ± 0.01 ^cde^	0.22 ± 0.01 ^bc^	2.31 ± 0.19 ^cd^	0.06 ± 0.01 ^b^
NB + Milk powder	0.28 ± 0.01 ^a^	0.25 ± 0.01 ^a^	3.50 ± 0.12 ^a^	0.07 ± 0.01 ^a^
SM + Milk powder	0.27 ± 0.01 ^abc^	0.24 ± 0.01 ^a^	2.81 ± 0.11 ^b^	0.08 ± 0.01 ^a^
NB + Whey protein concentrate	0.25 ± 0.01 ^bcd^	0.22 ± 0.01 ^bc^	2.32 ± 0.18 ^cd^	0.06 ± 0.01 ^bc^
SM + Whey protein concentrate	0.27 ± 0.01 ^ab^	0.22 ± 0.01 ^bc^	1.90 ± 0.11 ^de^	0.06 ± 0.01 ^b^
*p*-value	<0.001	<0.001	<0.001	<0.001

^a–f^ Means + standard errors (*n* = 18), followed by a different superscript in a given row, are significantly different (*p* < 0.05).

## Data Availability

The data presented in this study are available on request from the corresponding author. The data are not publicly available due to confidentiality agreement with the processing plant where samples were obtained.

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
