# Peer review of "Restoring Functionalities in Chicken Breast Fillets with Spaghetti Meat Myopathy by Using Dairy Proteins Gels"

_gels, 2022, doi:10.3390/gels8090558_

Round 1

Reviewer 1 Report

There is less experimental results to support this research, authors should add some experiment to deeply discuss your research aim., eg structural analysis, digestiable analysis. The English writing is too boring and unreadable, authors also should rephrase your paper carefully.

Author Response

Please see as below.

There is less experimental results to support this research, authors should add some experiment to deeply discuss your research aim., eg structural analysis, digestiable analysis. The English writing is too boring and unreadable, authors also should rephrase your paper carefully.

-We understand the suggestion of the reviewer. We have not conducted further experiments for this short communication. More complex studies could be considered as the next step of the study.

-Several paragraphs have been added for clarification and explanation.

Reviewer 2 Report

Restoring functionalities in chicken breast fillets with spaghetti meat myopathy by using dairy proteins gels

The idea is good and the organization and the structure of the manuscript are satisfactory. However there are major concerns should be considered before publication.

In the abstract: more details are required

Introduction section is too short. Try to improve it and add the hypothesis at the end of the introduction (before the objectives).

The material and methods section doesn't contain any reference? Also add the version of SAS. Add appropriate references under each subtitles in materials.

In table 1: Add the significance or p value in a new row.

Check the reference

Author Response

please see as below:

The idea is good and the organization and the structure of the manuscript are satisfactory. However there are major concerns should be considered before publication.

In the abstract: more details are required

  • Agreed, values of results were added. More details were also added to describe SM. A conclusive statement was added to the end of the abstract

Introduction section is too short. Try to improve it and add the hypothesis at the end of the introduction (before the objectives).

  • A paragraph has been added to the introduction to explain the formation of a strong gel using dairy in meat batters. Hypothesis added before objective of the study

The material and methods section doesn't contain any reference?

Also add the version of SAS.

               -added the version of SAS (9.4)

Add appropriate references under each subtitles in materials.

               -added references to each subsection of materials 

In table 1: Add the significance or p value in a new row.

-P-values added in a new row

Check the reference

               -doi numbers were added to references with missing doi

Reviewer 3 Report

Please add a sentence at the end of the introduction where you clarify the novelty of the work.

Why do the authors use the term "Spaghetti Meat" using all caps and not "spaghetti meat"?

Line 47. Also, WPI is an exception for the SM case.

Figures 1 and 2 and Table 1. The caption should identify the samples presented.

Line 66. Figure 2 should be mentioned in the text.

Conclusion. Please give an insight on the next steps for validating the proposed process and how it can be used by the industry.

Author Response

please see as below:

Please add a sentence at the end of the introduction where you clarify the novelty of the work.

  • A sentence was added to the end of the introduction, explaining that this short communication seeks to verify the effect of dairy protein additives on fillets with SM, featuring different proximate compositions compared to normal breast fillets.

Why do the authors use the term "Spaghetti Meat" using all caps and not "spaghetti meat"?

  • Agreed, changed to lower case.

Line 47. Also, WPI is an exception for the SM case.

  • Agreed, statement changed appropriately to represent the results

Figures 1 and 2 and Table 1. The caption should identify the samples presented.

  • Captions of figures and tables changed and identified the samples presented 

Line 66. Figure 2 should be mentioned in the text.

  • Agreed, figures 2 is now mentioned in the discussion.

Conclusion. Please give an insight on the next steps for validating the proposed process and how it can be used by the industry.

- A few sentences were added to suggest further studies and the application of the results of this study for the industry

Reviewer 4 Report

The manuscript focused on restoring functionalities in chicken breast with spaghetti meat include cooking loss and texture properties. However, the results are so simple and less of novelty. Some comments are as follows:

For the introduction, much more information about the attitude of many consumers towards the SM should be added.

For the figure1/2, the same color is not recommended between different groups, maybe the authors could set all the samples with dairy protein to black and all the samples without dairy protein to white.

For the discussion, the reasons why the addition dairy protein has an impact on its functionality should be further descriptions.

For the 4.2cooking and cooking loss, the different samples from 5℃ to internal temperature of 72 ℃ should be different, the 1.25h of heating time maybe too long for the chicken samples, how the heating time is selected?

For the references, the DOI of several references should be added, it is not rigorous.

For the functionality of samples, the photos of samples should be added, the sample with different functionalities have different color, surface form etc.

Author Response

Please see as below:

The manuscript focused on restoring functionalities in chicken breast with spaghetti meat include cooking loss and texture properties. However, the results are so simple and less of novelty. Some comments are as follows:

For the introduction, much more information about the attitude of many consumers towards the SM should be added.

               -a paragraph has been added to the introduction on the attitude of consumers towards

               SM myopathies

For the figure1/2, the same color is not recommended between different groups, maybe the authors could set all the samples with dairy protein to black and all the samples without dairy protein to white.

-figures 1,2 are overhauled. Now samples with similar proteins additives are in the same color, and bars with stripes represent samples with SM

For the discussion, the reasons why the addition dairy protein has an impact on its functionality should be further descriptions.

  • Details were added to explain about gel formation between dairy proteins and SM samples. Figure 3. Added to visualize the difference between SM samples with and without adding caseinate.

For the 4.2cooking and cooking loss, the different samples from 5℃ to internal temperature of 72 should be different, the 1.25h of heating time maybe too long for the chicken samples, how the heating℃  time is selected?

-the material and methods followed similar studies on dairy proteins and poultry meat batters. Because the samples were heated from refrigeration to 72℃ in the water bath with cold water, it took around 1.25h for the internal temperature to reach 72℃.

For the references, the DOI of several references should be added, it is not rigorous.

               -DOI added to references where applicable.

For the functionality of samples, the photos of samples should be added, the sample with different functionalities have different color, surface form etc.

  • A picture of samples with and without caseinate was added as Figure 3.

Round 2

Reviewer 1 Report

None

Author Response

Some revisions were made

Reviewer 4 Report

Yes, it has been improved. However, I advise some addtional reviewers are necessary.

Author Response

Some revisions were made.